# Age Estimate of *GJB2*-p.(Arg143Trp) Founder Variant in Hearing Impairment in Ghana, Suggests Multiple Independent Origins across Populations

**DOI:** 10.3390/biology11030476

**Published:** 2022-03-21

**Authors:** Elvis Twumasi Aboagye, Samuel Mawuli Adadey, Kevin Esoh, Mario Jonas, Carmen de Kock, Lucas Amenga-Etego, Gordon A. Awandare, Ambroise Wonkam

**Affiliations:** 1West African Centre for Cell Biology of Infectious Pathogens (WACCBIP), University of Ghana, Legon, Accra P.O. Box LG 54, Ghana; etaboagye@st.ug.edu.gh (E.T.A.); smadadey@st.ug.edu.gh (S.M.A.); lamengaetego@ug.edu.gh (L.A.-E.); gawandare@ug.edu.gh (G.A.A.); 2Division of Human Genetics, Faculty of Health Sciences, University of Cape Town, Cape Town 7925, South Africa; esohkevin4@gmail.com (K.E.); mario.jonas@uct.ac.za (M.J.); carmen.dekock@uct.ac.za (C.d.K.); 3McKusick-Nathans Institute and Department of Genetic Medicine, John Hopkins University School of Medicine, Baltimore, MD 21205, USA

**Keywords:** connexin 26 (Cx26), non-syndromic hearing impairment, *GJB2*-p.Arg143Trp (c.427C > T) founder variant, variant origin, Ghana

## Abstract

**Simple Summary:**

The incidence of the *GJB2*-p.(Arg143Trp) founder mutation in numerous populations of different ethnolinguistic and geographical backgrounds has generated interest on the provenance and age of the variant, which is predominantly associated with non-syndromic hearing impairment in Ghana. To interrogate the multiple possible independent origins of the *GJB2*-p.(Arg143Trp) variant, we estimated the age of the variant in Ghana by Bayesian inference, using linked makers in whole-exome sequencing data from hearing-impaired unrelated individuals who are homozygous for the *GJB2*-p.(Arg143Trp) variant, and nonaffected controls from the same population. We estimated the age of the variant as approximately 9625 years, from a common indigenous Ghanaian ancestor. Haplotype backgrounds in Ghanaian hearing-impaired individuals were apparently different from those of their Japanese counterparts. We observed low haplotype diversity in the *GJB2*-p.(Arg143Trp) genomic region for homozygous individuals compared to the normal hearing Ghanaian controls, although the recombination rate is relatively high in the region. The *GJB2*-p.(Arg143Trp)-positive individuals shared no common haplotype with the Ghanaian *GJB2*-p.(Arg143Trp)-negative controls, and as well as no common haplotype with data extracted from populations in 1000 Genomes, further supporting the multiple independent origins of *GJB2*-p.(Arg143Trp) in the global population.

**Abstract:**

Gap junction protein beta 2 (*GJB2*) (connexin 26) variants are commonly implicated in non-syndromic hearing impairment (NSHI). In Ghana, the *GJB2* variant p.(Arg143Trp) is the largest contributor to NSHI and has a reported prevalence of 25.9% in affected multiplex families. To date, in the African continent, *GJB2*-p.(Arg143Trp) has only been reported in Ghana. Using whole-exome sequencing data from 32 individuals from 16 families segregating NSHI, and 38 unrelated hearing controls with the same ethnolinguistic background, we investigated the date and origin of p.(Arg143Trp) in Ghana using linked markers. With a Bayesian linkage disequilibrium gene mapping method, we estimated *GJB2*-p.(Arg143Trp) to have originated about 9625 years (385 generations) ago in Ghana. A haplotype analysis comparing data extracted from Ghanaians and those from the 1000 Genomes project revealed that *GJB2*-p.(Arg143Trp) is carried on different haplotype backgrounds in Ghanaian and Japanese populations, as well as among populations of European ancestry, lending further support to the multiple independent origins of the variant. In addition, we found substantial haplotype conservation in the genetic background of Ghanaian individuals with biallelic *GJB2*-p.(Arg143Trp) compared to the *GJB2*-p.(Arg143Trp)-negative group with normal hearing from Ghana, suggesting a strong evolutionary constraint in this genomic region in Ghanaian populations that are homozygous for *GJB2*-p.(Arg143Trp). The present study evaluates the age of *GJB2*-p.(Arg143Trp) at 9625 years and supports the multiple independent origins of this variant in the global population.

## 1. Introduction

Hearing impairment (HI) is the commonest sensory human defect, with an estimated 2.5 billion people predicted to develop some degree of hearing difficulty worldwide by 2050 [1]. Genetic factors have been implicated in 50–60% of congenital HI, of which 80% are non-syndromic [2,3]. Genetic heterogeneity in HI has been well-demonstrated, with about 124 genes identified to date [4]. *GJB2* mapped to the DFNB1 locus is the most common gene associated with HI globally [5].

Hearing impairment associated *GJB2* alleles, are highly population-specific, with *GJB2*-c.35delG (p.Gly12Valfs*2) common in populations from Europe and the Middle East [6]; c.235delC (p.Leu79Cysfs*3) in East Asia [7]; c.845G>A (p.Val37Ile) prevalent in South-East Asia [8]; and c.71G>A (p.Trp24^*^), c.167delT (p.Leu56Argfs*26), and c.427C>T (p.Arg143Trp) common in India [9], Ashkenazim Jews [10], and Ghana [11,12], respectively. Although the molecular basis underlying the increased carrier frequencies of these DFNB1 alleles (0.7–2.4%) across populations remains to be elucidated [13,14], a possible heterozygote advantage has been proposed. For example, the increased survival of immortalized NEB-1 keratinocyte and NIH 3T3 fibroblast cells for *GJB2*-c.35delG carriers has been reported [15]. Moreover, *GJB2*-p.(Arg143Trp) has been suggested to be associated with a thicker skin epidermis, and increased sodium and chloride ion concentrations in sweat of *GJB2*-p.(Arg143Trp) homozygous individuals in Ghana, with possibly related increased protection against insect bites, and provide an unfavorable osmotic milieu for pathogen invasion resistance to malaria [15].

The first observation alluding to multiple independent origins of *GJB2*-p.(Arg143Trp) came from a phylogenetic analysis of Y-chromosomes of populations reporting the variant where no plausible shared ancestry could be established among the populations [14]. In 2020, a haplotype analysis of *GJB2* mutations frequently reported in Japanese populations established that the *GJB2*-p.(Arg143Trp) variant evolved as the result of a founder effect. The study suggested that the variant may have occurred as multiple events, and its age was estimated in the Japanese population at 6500 years ago [16]. In Ghana, *GJB2*-p.(Arg143Trp) is the commonest NSHI-associated variant, with a reported prevalence of 25.9% in affected families and a carrier frequency of 1.4% in the general population [13]. Reports of a low-to-moderate prevalence of the *GJB2*-p.(Arg143Trp) variant in other populations [14] has raised questions as to whether it originated through multiple events in different populations, or whether it originated from a single African (Ghanaian) ancestor and subsequently spread to other populations through migration and/or gene flow. If *GJB2*-p.(Arg143Trp) evolved via multiple independent events, then its higher frequency in Ghana might reflect a much longer presence there. The social, educational, and economic circumstances of the deaf have greatly improved over the past 300–400 years, and this is thought to have contributed towards improved fitness among the deaf population [17]. Such improved fitness would increase the frequencies of HI gene variants. Nevertheless, such a high frequency could have also been driven by multiple other factors, including natural selection, population expansions, and genetic drift.

In this study, we took advantage of the available whole-exome sequencing (WES) data for Ghanaian families with hearing-impaired relatives to investigate the age of *GJB2*-p.(Arg143Trp) using a linkage disequilibrium-based disease mapping algorithm. We also performed haplotype analyses in Ghanaian populations, as well as in other continental populations extracted from the 1000 Genomes project. We show that the age of p.(Arg143Trp) in Ghana predates earlier reports, e.g., in Japan.

## 2. Materials and Methods

### 2.1. The Study Participants

The study was conducted following the Helsinki Declaration for the participants’ safety, and the protocol was approved by the Ethics Committee for Basic and Applied Science (ECBAS), University of Ghana (ECBAS 053/19-20).

Genomic DNA (gDNA) was isolated and purified from the peripheral blood samples of 32 NSHI affected individuals homozygous for the *GJB2*-p.(Arg143Trp) variant from 16 Ghanaian families, and 38 unrelated hearing controls negative for the variant and with no family history of deafness were selected from a large study investigating the genetic etiology of congenital hearing impairment in Ghana [13]. Participants responded to a well-designed structured questionnaire on family history, clinical records, self-reported ethnicity, and photographs of affected individuals taken for clinical confirmation to exclude suspected acquired and syndromic HI. Pedigrees were documented for at least three generations in recruited families. A hearing-impaired individual was defined as someone who is not able to hear well at the normal hearing threshold of 20 dB or better in one or both ears [18] and with no identified associated clinical condition. Audiological and clinical examinations were performed on all affected kindreds, and they were diagnosed as having profound NSHI (with an average tone over 91dB) using the KUDUwave Audiometer eMoyo, Radioear (Northcliff, Johannesburg, South Africa). Both verbal and signed consent were obtained from all participants or their parents/guardians before recruitment into the study, with guaranteed anonymity for strict scientific use of the samples and data generated.

### 2.2. Whole-Exome Sequencing (WES)

Whole-exome sequencing was performed at Omega Bioservices (Norcross, GA, USA) using a pair-end 150-bp run format on the Illumina HiSeq 2500 platform following the manufacturer’s instructions. The Nextera Rapid Capture Exome kit^®^ (Illumina, San Diego, CA, USA) was performed to prepare the libraries, and the resulting libraries were hybridized with a 37-Mb probe pool to enrich the exome sequences, as previously described [19]. The generated library sizes ranging from 300 bp to 421 bp were quantified by quantitative PCR (qPCR) before sequencing on an Illumina HiSeq 2500 sequencer Next-Generation Sequencing (NGS) platform to generate pair-end reads with 100 × depth of coverage.

### 2.3. Whole-Exome Sequence Variant Calling

Sequencing data were processed using Omega Bioservices (Norcross, GA, USA) developed pipelines on the Illumina DRAGEN Germline Pipeline v3.2.8. Briefly, high-quality reads were aligned to the human reference genome GRCh37/hg19 using DRAGEN software version 05.021.408.3.4.12. After sorting and duplicate marking, the base quality scores were recalibrated using the genome analysis toolkit (GATK v4.0.6.0) [20]. Variants were called using GATKHaplotypeCaller, and individual genomic variant call format (gvcf) files were generated. Joint variant calling was then performed using GATK GenotypeGVCF [21]. The sex of each individual was verified using Plink version 1.9 [21]. The familial relationships for all members were verified using the Kinship-based INference for Gwas (KING) algorithm [22].

### 2.4. Variant Filtration and Data Quality Control

The GATK variant quality score recalibration (VQSR) was performed on the genotype data, consisting of 291,189 variants. All biallelic single-nucleotide variants (SNVs) that “PASSed” all filters were then extracted into a new VCF file, extracting only those variants with a depth of coverage (DP) > 8 and genotype quality (GQ) > 20. The quality of the data was further ascertained by computing various quality metrics (i.e., average heterozygosity, singleton by sample, average depth, and the transition/transversion (Ts/Tv) ratio) using bcftools stats and plotted using plot-vcfstats [23].

### 2.5. Haplotype Estimation (Phasing) and Genotype Imputation

Haplotype estimation (phasing) was performed using BEAGLE v5.1 [24]. Phasing was performed first without imputation to produce an “unimputed” dataset and then with imputation to produce an “imputed” dataset in order to mimic the genomic data for age estimation. The phased unimputed data was then merged with 2504 samples from the 1000 Genomes project, phase 3, version 5 (1KGP3v5) dataset to produce a merged superset of 2574 samples using the bcftools *--isec* command, keeping only the variants that were present in both datasets. Further quality control was performed to exclude poor-quality variants by removing variants with minor allele frequency (MAF) < 1%, variants with genotype quality (GQ) < 90%, and variants that failed the Hardy–Weinberg equilibrium (HWE) test at a *p*-value threshold of 1 × 10^−6^. Additionally, for the imputed data dataset, all variants with an imputation accuracy less than 90% were excluded.

### 2.6. Sample Selection for Age Estimation

A set of 32 samples that were homozygous for the *GJB2*-p.(Arg143Trp) variant was separated into two sets of 16 unrelated (samples) each (hereafter referred to as R143W-1 and R143W-2, respectively). The KING software was then used to confirm that each set consisted only of unrelated samples. Briefly, identity-by-descent (IBD) reports were computed for each set separately using the *--ibdseq* command. In both datasets, two individuals from two families were inferred to be 4th-degree relatives, and one individual of the related pair was therefore excluded, leaving 15 unrelated samples for each set. Thirty-eight (38) unrelated samples that were negative for p.(Arg143Trp) (hereafter referred to as GHA-38) were then selected and merged with both R143W-1 and R143W-2 to produce two separate datasets of 53 samples each (hereafter R143W-1N and R143W-2N, respectively). KING software was again used to ascertain that each set consisted only of unrelated samples [22].

### 2.7. Markers Selection

The Plink1.9 *–r^2^* command was used to compute the linkage disequilibrium (LD) within a 2-Mb region on chromosome 13q12 surrounding the p.R143W variant (1 Mb on each side of the variant) using an LD threshold of 0.1 [22]. Four (4) SNVs (rs115802719, rs732021, rs17075877, and rs9578260) in LD with c.427C>T (p.(Arg143Trp)) were selected for the age estimation using the phased unimputed dataset. The allele frequencies of the variants were computed for the R143W-1 and R143W-2 and GHA-38 datasets separately. The allele frequencies of the variants for continental populations were extracted from the dbSNP database. In addition, a recombination plot of the 2-Mb region around p.(Arg143Trp) was generated from the HapMap recombination map [25].

### 2.8. GJB2: p. (Arg143Trp) (c.427C>T) Variant Age Estimation

The age of *GJB2*: p. (Arg143Trp) (c.427C>T) was estimated in the phased unimputed as well as the phased imputed datasets using a nextflow-based [26] workflow (https://github.com/esohkevin/mutationAge/tree/main) (Accessed 25 November 2021) that utilized the Disequilibrium Mapping Likelihood Estimation (DMLE + 2.3) algorithm [27]. DMLE + 2.3 allows a Bayesian inference of the allele age and location based on the LD detected in selected markers, given some population genetic and demographic parameters, including a genetic map (in centimorgans distance), target population average growth rate, and frequency of the mutation in the general population. The program applies the Markov Chain Monte Carlo (MCMC) algorithm to perform massive random simulations in order to estimate the posterior probability density of the occurrence of each event: mutation location, ancestral state, and allele frequency. In addition, it has a feature that allows multiple chains to be run simultaneously, thereby providing a means of assessing the robustness of the data and pipeline parameters by checking the convergence at the true mutation location if the location is set as unknown and the software is instructed to iterate over it. We therefore set the software to iterate over the mutation location in our data. Two chains were run for R143W-1N, R143W-2N, and R143W-1N-imputed each, setting 2,000,000 burnin and 1,000,000 main iterations. The allele frequency of the mutation in the general population was estimated at 0.00024 using the world population data (wordometer) [28]. The genetic map was computed (using a custom R script) by subtracting the distance of the first (5′-most) marker position from each subsequent position (hence, the first marker position was zero). The distances were then converted to centimorgans (cM) using the approximation 1 cM = 1,000,000 bp. The average population growth rate of 2.15% in Ghana as of 2020 according to The World Bank data was used [29,30].

### 2.9. Principal Component and Ancestry Analyses

The merged set of R143W-1N and 1KGP3v5 data was LD-pruned, such that all variant pairs with LD ≥ 0.2 within any 50-bp region using a window size of 10 SNVs were excluded. Five axes of genetic variations (principal components—PCs) were then computed using the PLINK2 *--pca* command [21], and a custom R script was used for visualization. For the global ancestry analysis, 50 samples from each of the 1KGP3v5 populations were randomly selected and merged with R143W-1N and LD-pruned as described above. The ancestral proportions were then computed using Admixture v1.3 [31]. Admixture utilizes a maximum likelihood approach to estimate the underlying ancestral coefficients and then a moving block bootstrap approach for estimating the standard errors. The analysis was done with 11 cross-validation runs (K = 1–11) and 300 bootstrap runs. The AncestryPainter perl program was used to visualize the ancestral proportions [32].

### 2.10. Haplotype Analysis

Haplotype frequencies were computed for GHA-38, R143W-1, and R143W-2 separately, based on the five markers that were selected for age estimation (including p.(Arg143Trp)) using the phased unimputed dataset. Haplotype frequencies were also computed for the R143W-1N + R143W-2N + GHA-38 + 1KGP3v5 merged superset based on the four markers that were in LD with p.R143W, since p.(Arg143Trp) was absent in the 1000 Genomes reference panel.

## 3. Results

### 3.1. Characteristics of Study Participants

From the clinical history and examinations, no environmental factors were attributed to the cause of the NSHI in the affected cohorts selected, and no evidence of neurologic (vertigo, dizziness, etc.) or ophthalmologic (distorted vision, photophobia, eye pain, etc.) symptoms were identified. The physical examinations also showed no vestibular or any systemic abnormalities in the affected individuals. At least a history of inter- and intragenerational NSHI phenotype variability was described in a majority of the unrelated affected families, with most affected kindreds having their first formal audiological assessment performed when it was a requirement for enrollment during school admission. Pedigrees and audiological assessments of representative familial cases showed bilateral profound sensorineural NSHI.

### 3.2. Features of Markers Selected for Age Estimation and Haplotype Analysis

Table 1 and Appendix A show the general features and quality metrics of the four (4) SNVs selected in the WES data in addition to p.(Arg143Trp) for age estimation and haplotype analysis based on LD within the 2-Mb region of p.(Arg143Trp) (c.427C>T), respectively. The four markers in the unimputed set occurred in the zinc finger MYM-type containing 2 gene (*ZMYM2*) and GJB2 genes (Figure 1), spanning a 128.4-Kb region. Eight SNVs that were in LD with p.(Arg143Trp) were selected for the age estimation in the imputed data (Appendix A). Notably, all the eight markers occurred in *GJB2* (spanning a 193-bp region), the majority of which were intronic variants (Figure 1). Apart from p.(Arg143Trp) (rs80338948), only one other marker (rs9578260) was present in both the unimputed and the imputed sets, and these two markers were present in the WES data, while the rest were imputed. In general, LD among the selected markers was low and short-ranged; this was lower for the unimputed data (range: 0.10–0.34) as compared to the imputed data (range: 0.36–0.38). The low LD around p.(Arg143Trp) was consistent with the relatively high recombination rate around the variant (especially upstream), as revealed by the recombination plot (Appendix A).

### 3.3. Origin of p.R143W in Ghana

The age of p.(Arg143Trp) was estimated using two different marker sets: a five-marker set in the unimputed data and a nine-marker set in the imputed data. There was virtually no difference in the estimates of the R143W-1N and R143W-2N datasets, and therefore, only the results of R143W-1N are herein presented. Using the five-marker unimputed set, the two analysis chains converged at the correct mutation location (0.12798) less than 500,000 iterations into the burnin iterations and remained stable throughout (Appendix A), simultaneously estimating the correct mutation location (Figure 2A) while also estimating the mutation age (Appendix A). Figure 2b shows the posterior density of the mutation age with a mode of roughly 385 generations (9625 years) and a 95% credible set of ages ranging from 335 to 500 generations (8375–12,500 years), giving a generation time of 25 years. Using the nine-marker imputed set, the two chains converged at the correct location (0.000365; Appendix A) approximately 1,800,000 iterations into the burnin iterations, while the age of the mutation was estimated at roughly 380 generations (~9500 years) ago, with a credible set of ages between 315 and 500 generations (7875–12,500 years) (Appendix A). Apparently, the age and location estimates were more stable for the unimputed dataset as compared to the imputed set, likely reflecting the effect of even a small fraction of the imputation error on the age estimation algorithm. However, the algorithm still equilibrated before the main (good) iterations started, implying that the tool retains the power to estimate the mutation age given that higher values of the burnin iterations are set in order to allow for enough equilibration time. 

### 3.4. Haplotype and Ancestry Analyses Confirm Independent Origin of p.(Arg143Trp) in Ghana

To further understand how p.(Arg143Trp) may have evolved in Ghana, thirteen (13) unique haplotypes were generated for R143W-1, R143W-2, and GHA-38 separately based on the five-marker set, and nine unique haplotypes were generated for Ghanaian populations and the 1KHP3v5 merged superset, and the haplotype frequencies were compared. Figure 3A shows the haplotype distribution among the different Ghanaian sample sets. The R143W-1 and R143W-2 populations exhibited a much lower haplotype diversity in the p.(Arg143Trp) genomic region compared to the GHA-38 (p.(Arg143Trp)-negative) population, possibly reflecting some evolutionary constraints in the region in the genetic background of homozygous p.(Arg143Trp). In addition, the p.(Arg143Trp)-positive samples were clearly distinguished from the GHA-38 samples, as there were no shared haplotypes among the two sample sets (Appendix A). Interestingly, four (4) of the six (6) haplotypes that carried p.(Arg143Trp) were also homozygous for the rs9578260 intron variant, which has been reported in hearing-impaired cases and classified as benign on ClinVar (https://www.ncbi.nlm.nih.gov/snp/rs9578260#clinicalsignificance) (Accessed 15 December 2021). That is, over 90% of all Ghanaian p.R143W homozygous cases are also homozygous for the *GJB2* intron variant rs9578260. Although rs9578260 is also highly prevalent in p.(Arg143Trp)-negative African populations (MAF = 0.22), it is absent in Asia and extremely rare in Europe. This clearly indicates that p.(Arg143Trp) is carried on different haplotype backgrounds in Ghanaian and Japanese populations, as well as European populations. Moreover, a majority of the markers that were selected for allele age estimation based on LD with p.(Arg143Trp) are either absent or rare in Asia and Europe (Table 1), further supporting independent evolutions of p.(Arg143Trp) in the different ancestries. A striking observation was that the haplotypes generated from the four selected markers were strongly ancestry informative, given that they resolved all the populations in the Ghanaian and 1000 Genomes set at the continental and subcontinental levels (Figure 3b). They revealed a higher haplotype diversity in African populations (as expected), as well as an admixture in American populations (PUR, CLM, MXL, and PEL). These observations were consistent with the PCA (Figure 3c) and ancestry analysis (Figure 3d). The ancestry analysis at k = 5 resolved all five continental populations in the 1000 Genomes project (AFR, AMR, EUR, SAS, and EAS) (Appendix A), while the Ghanaian populations clustered with African populations in close proximity to the Mende population from Sierra Leone (MSL) and the Yoruba from Nigeria (YRI). Increasing the k-parameter (number of subpopulations) from k = 8 up to k = 11 in order to resolve the populations further revealed no shared genetic diversity among the Ghanaian and Japanese populations (Appendix A), meaning that it is highly unlikely to find shared haplotypes among the populations, further lending support to the notion of an independent origin of p.(Arg143Trp) in Ghana.

## 4. Discussion

The present report shows that the *GJB2*-p.(Arg143Trp) founder variant has been present for approximately ~9625 years in Ghana and most likely evolved independently from its occurrence in Japan [14,16]. This suggests that Ghanaian families with the p.R143W variant are likely descendants of an ancestor who lived some ~385 generations ago. The finding elucidates the origin of the founder variant segregating in families with NSHI in Ghana [33,34,35].

The much lower haplotype diversity observed in the p.(Arg143Trp) genomic region in p.(Arg143Trp) homozygotes compared to hearing controls (p.(Arg143Trp)-negative) might reflect an evolutionary constraint in the genomic region due to one or more factors, including natural selection, and/or a higher inbreeding than would be expected under neutrality due to assortative mating among the deaf population. The chromosomal region around *GJB2* is known to contain high recombination rates [36] and harbor recombinational hotspots, which may explain the high mutability of the *GJB2* gene evident in the reported population-specific *GJB2* variants worldwide [14]. This would further support the notion of multiple independent mutational events of p.(Arg143Trp), and other *GJB2* gene variants in different populations [37,38].

The factors that may have contributed to maintaining the *GJB2*-p.(Arg143Trp) deleterious allele at an appreciable frequency in Ghana include an unusually higher rate of positive assortative mating in deaf communities, and a probable high rate of consanguinity among the early carriers of the variant [11,12,39]. The observed SNVs (in the five-marker haplotype sets) specific to the p.(Arg143Trp) homozygotes and absent in GHA-38 (*GJB2*-p.(Arg143Trp)-negative) individuals could support the consanguineous assertion. The impact of the geographical location, size, and the isolated nature of the early population linked with the variant in Ghana coupled to the high rate of non-migration among the settlers of the village, fueled by fear of discrimination, stigmatization, and societal neglect, may also explain the high prevalence of the *GJB2*-p.(Arg143Trp) founder variant in this region as compared to elsewhere [5,6,14,37]. Furthermore, the current finding is consistent with reports of the combined impact of improved fitness [40], unknown genetic modifiers, and noncomplementary marriages following sign language introduction that have been implicated in the substantial increase of deafness alleles and could be extend to the perpetuation of the p.(Arg143Trp) alleles in the families investigated [41].

A heterozygote advantage in genetic conditions like hemochromatosis, sickle cell disease (SCD), and factor V Leiden thrombophilia has been implicated as the driver of the continued segregation of deleterious gene variants in populations. Heterozygote carriers of homeostatic iron regulator (*HFE*): p.(Cys282Tyr) in hemochromatosis are reported to have some protection from iron deficiency [42,43,44]. Sickle cell hemoglobin carriers (HbAS) hardly develop severe malaria episodes [45], and factor V Leiden thrombophilia variant carriers are also reported to be resistant to some bloodstream bacterial infections [46]. Similarly, *in vivo* and *in vitro* studies have demonstrated skin protection in carriers of the *GJB2*-c.35delG founder variant, thus implicating natural selection as driving the evolution of the variant [47]. Nevertheless, the lack of skin disorders in *GJB2*-associated NSHI cases means that the loss or gain of connexin gene function alone may not impact the epidermis development and function as alluded [48]. Although our haplotype analysis around the locus confirms it evolved independently, and the allele frequency is higher than would be expected under neutrality for a pathogenic variant, to date, the reported discrepancies in the distribution of the variant in NSHI cases are yet to be correlated with any past or present defined endemic condition(s) in the history of the populations reported with the variant.

The current *GJB2*-p.(Arg143Trp) age estimate falls within the period of Holocene climatic changes ~11,650 years ago (around the same time as the emergence of the *GJB2*- c.35delG variant in Anatolia ~10,000 years ago) that modulated the Neolithic movements and transitioned the natural ecosystem to pastoralism [49]. Archaeological and historical evidence of unfavorable climates in Africa between 7000 and 12,000 years ago show a significant increase in the number of lakes and ponds, breeding pathogenic vectors [50,51]. Consequently, human domestication of plants and animals in the subregion and exposure to these conditions may have increased contact with infectious pathogens like the malaria parasite that may have elicited the selection of certain pathogenic variants such as the sickle cell mutation, and possibly *GJB2*-p.(Arg143Trp) to protect against the deadly malaria parasite through a heterozygote advantage [52,53]. Therefore, the question of whether *GJB2*-p.(Arg143Trp) has coevolved with malaria deserves further scrutiny with genome-wide data, since connexin mutations causing syndromic sensorineural HI associated with epidermal disorders are coupled with additional variants, and the synergistic effect may be responsible for the observed epidermis differentiation [54,55]. Preferably, other neighboring African populations with shared geographical and ecological conditions and in whom the mutation has not been explored should also be sampled [56].

The little shared ancestry among the GHA and JPT populations, evidenced by haplotypes in Ghanaian individuals that were absent in the Japanese individuals and *vice versa*, clearly indicates that the two populations have different LD patterns and haplotype structures in the 2-Mb region around the p.(Arg143Trp) founder locus. This supports the independent evolution of *GJB2*-p.(Arg143Trp) in the two populations. Moreover, the absence of the p.(Arg143Trp) tag variant rs9578260 in Japanese populations, whereas it is highly prevalent in African populations, further suggests that *GJB2*-p.(Arg143Trp) is carried on different haplotype backgrounds in both populations and therefore may have highly likely evolved independently in these populations. The four-marker haplotypes were strongly ancestry informative and resolved populations in Ghana and the 1000 Genomes set at the continental and subcontinental levels. The unique haplotype proportions in *GJB2*-p.(Arg143Trp) homozygotes compared to the hearing controls may allow the use of the four tagged SNVs as surrogate secondary markers for the *GJB2*-p.(Arg143Trp) pathogenic variant in Ghana. Thus, this study shed light on the origin of the most common *GJB2*-p.(Arg143Trp) variant associated with the most NSHI in Ghana. Yet, how the *GJB2*-p.(Arg143Trp) variant has been maintained at appreciable frequencies in the Ghanaian population remains a puzzle that would most likely be delineated using whole-genome data from Ghanaian HI, as well as hearing populations.

A global haplotype analysis with the disease allele would have been ideal to provide haplotype blocks of the evolved allele with other markers in LD with it across populations. However, the absence of the disease mutation in the 1KGP3v5 database restricted the haplotype analysis to the use of four tagged markers that together form strong ancestry informative haplotypes. In addition to the absence of accurate population records, the study used raw observed population growth rates without taking into account the past population demographic history and the possibility of natural selection, which might influence the growth rates for the age estimation.

## 5. Conclusions

We demonstrated in this study the unique ancestral haplotype background in *GJB2*-p.(Arg143Trp) homozygotes individuals, confirming no significant shared genetic clusters between Ghana (sSA) ancestry and the highly homogenous Asian populations. The carriers perpetuating the deleterious allele (*GJB2*-p.(Arg143Trp)) in Ghana are descendants of an indigenous ancestor some ~385 generations ago. The high recombinational events around the disease locus may explain the shortening of the haplotype and possible balance selection evident in the relatively low LD observed in the mutated chromosomes. This study strongly supports that the *GJB2*-p.(Arg143Trp) variant evolved independently and segregated in the affected families, preserved by a selective mating pattern and consanguinity within the populations.

## Figures and Tables

**Figure 1 biology-11-00476-f001:**
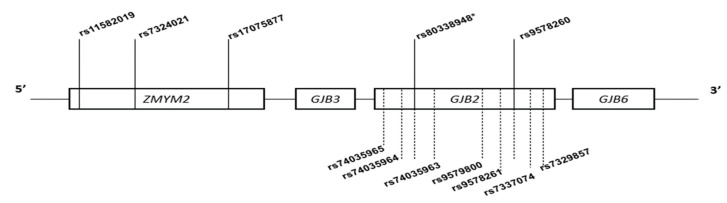
Schematic illustration of the five ancestral informative markers (AIMs) on the chromosome 13q12 locus (2-Mb region around the disease mutation) used for the age estimation and haplotype analysis in Ghanaian populations. Tagged SNVs in linkage disequilibrium were identified within a 128.445-kb region spanning the *GJB2*: c.427C>T (Arg.R143Trp) locus. One marker in *GJB2*: c.-22-12C>G and three markers in the zinc finger MYM-type containing 2 gene (*ZMYM2*): c.3345A>G, c.3037+27T>G, and c.2857A>G, respectively. Unimputed markers (5) are shown above, and imputed markers (7) are shown below, represented by dotted lines. * = disease allele.

**Figure 2 biology-11-00476-f002:**
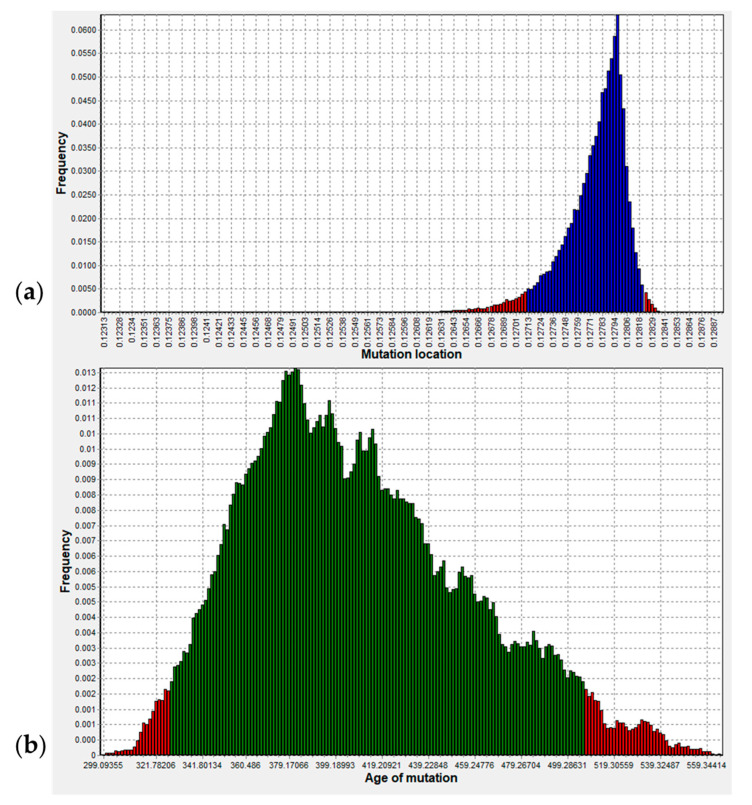
DMLE +2.3 estimated age of p.(Arg143Trp) (c.427C>T) in generations showing a 95% confidence interval (CI) after two chains: 2,000,000 burnin and 1,000,000 main iterations. (**a**) p.(Arg143Trp) mutation location and a 95% CI. (**b**) p.(Arg143Trp) (c.427C>T) mutation age and a 95% CI for the posterior probability density (the region in green).

**Figure 3 biology-11-00476-f003:**
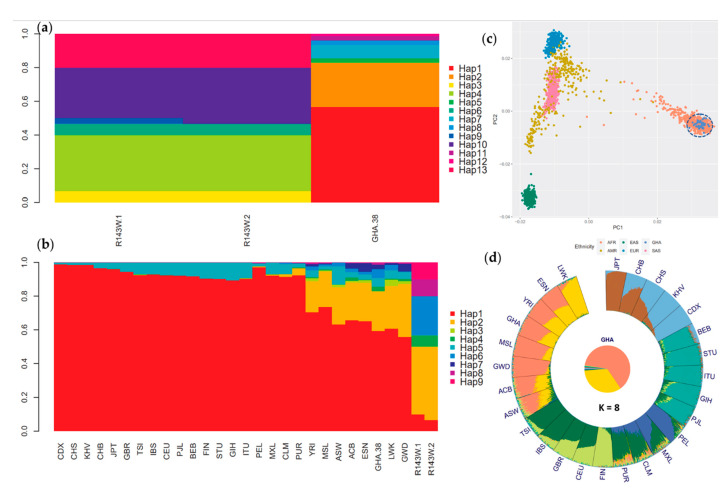
Haplotype frequencies in the R143W-1, R143W-2, GHA-38, and 1KGP3v5 populations and a global ancestry analysis. (**a**) Haplotype frequencies in Ghanaian samples R143W-1, R143W-2, and GHA-38 based on the 5-marker haplotypes. (**b**) Haplotype frequencies in Ghanaian (R143W-1, R143W-2, and GHA-38) and 1KGP3v5 populations based on the 9-marker haplotypes. (**c**) Principal Component Analysis showing the clustering of the continental populations. Ghanaian (GHA) population clustered with the African population as expected. EAS = East Asian Ancestry, AMR = American Ancestry, EUR = European Ancestry, and SAS = South Asian. (**d**) Bar plot of an unsupervised ADMIXTURE analysis of Ghanaian and the 1000 Genomes populations assuming 8 subpopulations (K). Luhya in Webuye, Kenya (LWK), Esan in Nigeria (ESN), Yoruba in Ibadan, Nigeria (YRI), R143W1-N Ghanaians (GHA), Mende in Sierra Leone (MSL), Gambian in Western Division—Mandinka (GWD), African Caribbean in Barbados (ACB), African Ancestry in SW USA (ASW), Toscani in Italia (TSI), Iberian Populations in Spain (IBS), Utah residents (CEPH) with Northern and Western European ancestry (CEU), British from England and Scotland (GBR), Finnish in Finland (FIN), Puerto Rican in Puerto Rico (PUR), Colombian in Medellín, Colombia (CLM), Mexican ancestry in Los Angeles CA USA (MXL), Peruvian in Lima Peru (PEL), Punjabi in Lahore, Pakistan (PJL), Gujarati Indians in Houston, Texas, USA (GIH), Indian Telugu in the UK (ITU), Sri Lankan Tamil in the UK (STU), Bengali in Bangladesh (BEB), Kinh in Ho Chi Minh City, Vietnam (KHV), Chinese Dai in Xishuangbanna, China (CDX), Han Chinese South (CHS), Han Chinese in Beijing, China (CHB), and Japanese in Tokyo, Japan (JPT).

**Table 1 biology-11-00476-t001:** Features of linkage disequilibrium-selected variants for the age estimate and haplotype analysis.

Marker	Chr:Pos	Ref/Alt	Gene	LD (*r^2^*)	Genetic Map (cM)	Alternate Allele Frequency (AAF)							
						R143W-1N	R143W-2N	GHA-38	AFR	EUR	AMR	Asian	EAS
rs115802719	13:20635310	A/G	*ZMYM2*	0.102	0.00000	0.1875	0.1875	0.0375	0.0253	0.000088	0.0256	0.0000	0.0000
rs7324021	13:20637140	T/G	*ZMYM2*	0.220	0.00183	0.3438	0.3438	0.1125	0.0705	0.05039	0.0700	0.0100	0.0110
rs1705877	13:20641422	A/G	*ZMYM2*	0.105	0.00611	0.2500	0.2500	0.0625	0.1595	0.12264	0.1600	0.1430	0.1600
*rs80338948	13:20763294	G/A	*GJB2*	1.000	0.12798	1.0000 *	1.0000 *	0.0000 *	0.0002	0.000024	0.0002	0.0002	0.0002
rs9578260	13:20763754	G/A	*GJB2*	0.341	0.12844	0.9062	0.9375	0.3250	0.2167	0.00082	0.2165	0.0000	0.0000

AFR = Africa; EUR = Europe; AMR = African Americans; Asian = Asia; EAS = East Asia; RSID = reference SNP ID; Chr:Pos = Chromosome and base pair position; Ref/Alt = Reference and alternate alleles; LD = linkage disequilibrium; cM = Centimorgan; AAF = alternate allele frequency; * = disease allele.

## Data Availability

Not applicable.

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
