# Peer review of "Age Estimate of GJB2-p.(Arg143Trp) Founder Variant in Hearing Impairment in Ghana, Suggests Multiple Independent Origins across Populations"

_biology, 2022, doi:10.3390/biology11030476_

Round 1

Reviewer 1 Report

Reviewer's report

The manuscript “Age Estimates of GJB2 (Connexin 26) p.R143W Founder Variant in Hearing Impairment in Ghana, suggest multiple independent origin across populations” by Aboagye et al reports the novel interesting data on natural history of a specific GJB2 founder variant p.(Arg143Trp) in population of African ancestry. This study aims to elucidate the origin and age of p.(Arg143Trp) in Ghana. To date, in the African continent, this variant has been only reported in Ghana where it is predominant in the GJB2-related hearing loss in Ghanaian patients. The age of p.(Arg143Trp) in Ghana was estimated by the Disequilibrium Mapping Likelihood Estimation (DMLE+2.3) algorithm to be 9,625 years (~ 385 generations ago). The authors also present the results of haplotype analysis comparing data extracted from Ghanaian individuals and those from the 1000 Genomes project. This analysis revealed different haplotype backgrounds for p.(Arg143Trp) in Ghanaian and this variant reported in Japanese, as well as among populations of European ancestry. All data obtained support the multiple independent origins of variant p.(Arg143Trp).

Methods are well described. Discussion and Conclusion sections are well written and seem to cover all the aspects of the study. Additionally, this study is of special interest since it was performed on one of populations of African ancestry which are characterized by very specific genetic contributors to hearing loss.

I believe this manuscript needs a minor revision and I have provided comments/suggestions for authors that I hope will help improve the manuscript.

My comments

1) All gene variants mentioned in the paper should be named according to the accepted guidelines using three letter codes for all, for example p.(Arg143Trp).

2) Page 13, lines 417-418: the references [36,37] do not match the statement “The chromosomal region around GJB2 is known to contain high recombination rates [36,37]…”

3) Unfortunately, in the text, I could not find exact information about the physical size of the haplotype carrying p.(Arg143Trp) in Ghanaian patients.

4) The manuscript is not free from typos and errors in English. For example, same of them:

  • Page 1, line 18: Incidence of the GJB2-p.R143W founder mutation numerous populations… - Incidence of the GJB2-p.R143W founder mutation in numerous populations…
  • Page 2, lines 49-50: The present study evaluate the GJB2-p.R143W to 9,625 years, and support the multiple independent origin of that variants multiple global population. - The present study evaluates the age of the GJB2-p.R143W to 9,625 years, and supports the multiple independent origin of that variants multiple global population. this variant in multiple global populations.
  • Page 2, line 57: Genetic – factors… - Genetic factors…
  • Page 3, line 104: Genomic DNA (gDNA) isolated and purified from the peripheral blood samples… -  Genomic DNA (gDNA) was isolated and purified from the peripheral blood samples… 
  • Page 3, lines 121-123: Whole exome sequencing was performed at Omega Bioservices (Norcross, GA, 121 USA), using pair-end 150 bp run format on Illumina HiSeq 2500 platform. Following the manufacturer’s instructions. -  Whole exome sequencing was performed at Omega Bioservices (Norcross, GA, 121 USA), using pair-end 150 bp run format on Illumina HiSeq 2500 platform following the manufacturer’s instructions.

Author Response

We are very grateful for the positive comments, the insightful suggestions, and editing that have greatly improved our manuscript.

Comment 1

All gene variants mentioned in the paper should be named according to the accepted guidelines using three letter codes for all, for example p.(Arg143Trp).

Authors' response

All gene variants are now revised to the three letter codes, as follows:

i. p.R143W changed to p.(Arg143Trp)

ii.   GJB2-c.35delG changed to GJB2-c.35delG (p.Gly12Valfs*2)

iii.  c.235delC changed to c.235delC (p.Leu79Cysfs*3)

iv.  p.V37I changed to c.845G>A (p.Val37Ile)

v. p.W24X, c,167delT and c.427T>C changed to c.71G>A (p.Trp24*), c.167delT (p.Leu56Argfs*26), and c.427C>T (p.Arg143Trp)

vi. p.(C282Y) changed to p.(Cys282Tyr)

Comment 2

Page 13, lines 417-418: the references [36,37] do not match the statement “The chromosomal region around GJB2 is known to contain high recombination rates [36,37]…”

Authors' response

Reference changed, now reads; The chromosomal region around GJB2 is known to contain high recombination rates [36] and …

Comment 3

Unfortunately, in the text, I could not find exact information about the physical size of the haplotype carrying p.(Arg143Trp) in Ghanaian patients.

Authors' response

The size of the region from which they were retrieved has been indicated in the main text as follows:

The four markers in the unimputed set occurred in the zinc finger MYM-type containing 2 gene (ZMYM2) and GJB2 genes (Figure 1), spanning a 128.4Kb region. Eight SNVs that were in LD with p.(Arg143Trp) were selected for age estimation in the imputed data (Table S2). Notably, all the 8 markers occurred in GJB2 (spanning a 193bp region), majority of which were intronic variants (Figure 1).

The haplotypes were defined by the occurrence or absence of mutation at each SNP position, and only homozygotes for each position were considered in each population.

Comment 4

Page 1, line 18: Incidence of the GJB2-p.R143W founder mutation numerous populations… - Incidence of the GJB2-p.R143W founder mutation in numerous populations…

Authors' response

Revised: now reads …. founder mutation in numerous populations of different ethnolinguistic...

Comment 5

Page 2, lines 49-50: The present study evaluate the GJB2-p.R143W to 9,625 years, and support the multiple independent origin of that variants multiple global population. - The present study evaluates the age of the GJB2-p.R143W to 9,625 years, and supports the multiple independent origin of that variants multiple global population. this variant in multiple global populations.

Authors' response

Revised: now reads …… suggesting a strong evolutionary constraint in this genomic region in Ghanaian population that are GJB2- p.(Arg143Trp) homozygous. The present study evaluates the age of GJB2-p.(Arg143Trp) to 9,625 years, and supports the multiple independent origin of this variant in multiple global populations

Comment 6

Page 2, line 57: Genetic – factors… - Genetic factors…

Authors' response

Revised, now reads; Genetic  factors have been

Comment 7

Page 3, line 104: Genomic DNA (gDNA) isolated and purified from the peripheral blood samples… -  Genomic DNA (gDNA) was isolated and purified from the peripheral blood samples… 

authors' response

Revised, now reads; Genomic DNA (gDNA) was isolated and purified from the peripheral.

Comment 8

Page 3, lines 121-123: Whole exome sequencing was performed at Omega Bioservices (Norcross, GA, 121 USA), using pair-end 150 bp run format on Illumina HiSeq 2500 platform. Following the manufacturer’s instructions. -  Whole exome sequencing was performed at Omega Bioservices (Norcross, GA, 121 USA), using pair-end 150 bp run format on Illumina HiSeq 2500 platform following the manufacturer’s instructions.

Authors' response

Revised, now reads; Whole exome sequencing was performed at Omega Bioservices (Norcross, GA, USA), using pair-end 150 bp run format on Illumina HiSeq 2500 platform following the manufacturer’s instructions.

Reviewer 2 Report

Manuscript ID: Biology-1631172

Title: Age estimates of GJB2 (Connexin 26) p.R143W founder variant in hearing Impairment in Ghana, suggest multiple independent origin across populations

First Author: Elvis Twumasi Aboagye

The paper reports on haplotype diversity in the gap junction protein beta 2 (GJB2) p.R143W variant genomic region in Ghanaian, Japanese and among populations of European ancestry. This variant is the largest contributor to non-syndromic hearing impairment in Ghana, with a prevalence of 25.9% in affected multiplex families. Using whole exome sequencing data from affected individuals, and unrelated normal hearing controls with the same ethnolinguistic background and with a Bayesian linkage disequilibrium gene mapping method, the authors estimated GJB2-p.R143W to have originated about 9,625 years ago from a common indigenous Ghanaian ancestor. Haplotype analysis comparing data extracted from Ghanaian and those from the 1000 Genomes project revealed that GJB2- p.R143W is carried on different haplotype backgrounds in Ghanaian and same founder variant reported in Japanese, as well as among populations of European ancestry, providing further support to the multiple independent origins of the variant.

The study is interesting and well designed. It provides evidence that segregation of GJB2-p.R143W variant in the affected families is preserved by a selective mating pattern and consanguinity within the populations.

Minor points:

Please remove (Connexin 26) from title and introduce it in the text (in the abstract for instance).

Please use the correct mutation nomenclature for the mutations cited in the text:

p.V37I, p.W24X, p.R143W should be replaced with p.(Val37Ile), p.(Trp24*), p.(Arg143Trp) respectively.

Author Response

We are very grateful for the revision and positive comments.

Comment 1

All gene variants mentioned in the paper should be named according to the accepted guidelines using three letter codes for all, for example p.(Arg143Trp).

Authors' responses

All gene variants revised to the three letter codes.

  1. R143W changed to p.(Arg143Trp)
  2. GJB2-c.35delG changed to GJB2-c.35delG (p.Gly12Valfs*2)
  3. 235delC changed to c.235delC (p.Leu79Cysfs*3)
  4. V37I changed to c.845G>A (p.Val37Ile)
  5. W24X, c,167delT and c.427T>C changed to c.71G>A (p.Trp24*), c.167delT (p.Leu56Argfs*26), and c.427C>T (p.Arg143Trp)
  6. p.(C282Y) changed to p.(Cys282Tyr)

Comment 2

Please remove (Connexin 26) from title and introduce it in the text (in the abstract for instance).

Authors' response

Revised, the title now reads:

Age estimates of GJB2-p.(ArgR143Trp) Founder Variant in Hearing Impairment in Ghana, suggests multiple independent origin across populations